# Coronary Arteries Aneurysms: A Case-Based Literature Review

**DOI:** 10.3390/diagnostics12102534

**Published:** 2022-10-19

**Authors:** Giuseppe Vadalà, Leandro Di Caccamo, Chiara Alaimo, Luca Di Fazio, Giovanni Ferraiuoli, Giancarlo Buccheri, Vincenzo Sucato, Alfredo Ruggero Galassi

**Affiliations:** 1Division of Cardiology, University Hospital Paolo Giaccone, Via del Vespro 129, 90100 Palermo, Italy; 2Department of Health Promotion, Mother and Child Care, Internal Medicine and Medical Specialties (ProMISE), University of Palermo, 90100 Palermo, Italy

**Keywords:** coronary artery aneurysm, giant coronary artery aneurysm, percutaneous coronary intervention, cardiac surgery

## Abstract

Coronary artery aneurysm (CAA) is an abnormal dilatation of a coronary artery segment; those coronary artery aneurysms that are very large in size are defined as giant. However, a standardized dimension cut-off to define giant CAAs is still missing. The reported prevalence of coronary aneurysms in the population who underwent coronary angiography ranges from 0.3% to 5%, and often CAAs are found in patient with aneurysms in other sites, such as the ascending or abdominal aorta. In half of the cases an atherosclerotic etiology could be recognized; often, CAA is found in the context of acute coronary syndrome. Seldomly, CAA is found at the autopsy of patients who died due to sudden cardiac death. Currently, very few data exist about CAA management and their prognostic relevance; moreover, CAA treatment is still not clearly codified, but rather case-based. Indeed, currently there are no published dedicated studies exploring the best medical therapy, i.e., with antiplatelets or anticoagulant agents rather than an interventional approach such as an endovascular or surgical technique. In this review, through two clinical cases, the current evidence regarding diagnostic tools and treatment options of CAAs will be described.

## 1. Introduction

**Definition.** Coronary artery aneurysm (CAA) is an abnormal dilatation of a coronary artery segment, 50% larger in diameter than the adjacent normal segments or the diameter of the patient’s largest coronary vessel [1,2].

When a CAA is very large in size it is defined as a giant coronary artery aneurysm (GCAA); however, there is still no clear consensus on the appropriate cut-off that must be used to defined GCAA. Indeed, diameters greater than 20, 40 or 50 mm, or quadruple the reference vessel diameter have all been proposed as cut-offs by different authors. Furthermore, in case of Kawasaki disease, it was proposed to define CAA as giant if its diameter is larger than 8 cm [3,4]. Conversely, the coronary ectasia has been well characterized [5,6].

**Epidemiology.** The reported prevalence of coronary aneurysms in the population who underwent coronary angiography ranges from 0.3% to 5%; this finding is more frequent in men than in women (83% vs. 60%) [3,7]. The right coronary artery is the vessel involved most frequently, followed by the left anterior descending artery [2,4]. Interestingly, in roughly 16% of patients who underwent coronary angiography, an association between CAA and aneurysms of the ascending or abdominal aorta was found [8,9].

**Etiology.** Half of CAAs have an atherosclerotic etiology, while Kawasaki’s disease, congenital anomalies, autoimmune diseases, Marfan’s syndrome and aneurysm of coronary artero-venous fistula (CAVF) are other possible etiologies. Moreover, mycotic, post-traumatic or iatrogenic have been described as alternative but quite rare etiologies [3,10,11].

## 2. Pathogenesis

The mechanisms leading to CAA formation are poorly understood. In those cases of aneurysms located close to coronary stenosis, a possible pathophysiological mechanism might be related to the turbulent blood flow produced by the stenosis itself. Indeed, such turbulence leads increased vessel wall shear stress close to the stenosis with consequent weakening and vessel dilatation [12]. Another hypothesis is that an intimal plaque rupture might result in ulceration of an already-degenerated middle tunic, thus favoring vessel dilatation [13].

Another possible pathogenesis for CAA is the dysfunction of the coronary microcirculation; indeed, as previously reported, coronary imaging, both intravascular ultrasound or optical coherence tomography, could help identify pathological coronary wall features which are not visible on coronary angiography, and could explain the microvascular dysfunction [14,15] However, whether microvascular dysfunction might lead to CAA development or vice versa remains unclear.

Donati et al., in a study with a population of 144 patients with coronary artery ectasia (CAE) and CA, reported high rates of myocardial infarction with non-obstructive coronary arteries (MINOCA) and ischemia with non-obstructive coronary arteries (INOCA), respectively, in 31% of the entire cohort of patients admitted to the hospital for acute coronary syndrome (ACS) and 42% of the patients complaining of stable coronary artery disease (SCAD) [16].

Based on these findings, further dedicated studies are required to definitively establish if the microcirculation dysfunction play a role in CAA development.

Finally, infections can, in rare cases, lead to the formation of so-called “mycotic coronary aneurysms”, described, i.e., after infective endocarditis, percutaneous coronary interventions or cardiac surgery such as coronary artery bypass, Bentall’s procedure and abdominal aortic aneurysm [AAA] repair. In most of the cases, these are due to gram-positive bacteria [17].

## 3. Diagnosis

CAA is often an incidental finding that occurs in patients undergoing coronary angiography in the context of acute coronary syndrome (ACS) or stable angina, or in those patients scheduled for heart valve surgery. Sometimes, its clinical presentation could be a sudden cardiac death due to aneurysm acute thrombosis, aneurysm rupture or distal thrombo-embolization [13].

Even if the coronary angiography is the gold standard technique for CAA diagnosis, the multi-slice computed tomography angiography (MSCTA) is being used more and more often because it provides supplementary information on vessel walls, the true aneurysm dimension, the possible thrombotic content and the aneurysm’s relationship with the neighboring structures [3]. Intravascular imaging such as intravascular ultrasound (IVUS) or optical coherence tomography (OCT) allows a more detailed characterization of coronary aneurysms. For example, these techniques can differentiate between true and false aneurysms and may show the morphologic changes underlying CAA development [18,19,20,21].

As an alternative, CAA might be evaluated by transthoracic echocardiography, cardiac magnetic resonance (CMR) imaging, and CMR angiography [4]. These imaging modalities are less invasive than coronary angiography and avoid patient exposure to ionizing radiations. This aspect is most important in cases of congenital or Kawasaki syndrome-related aneurism found in the pediatric population, both of which require a systematic follow-up of aneurysm’s dimensions [22].

## 4. Treatment

CAA therapy basically consists of conservative management by optimized medical therapy (OMT) or invasive treatment in selected cases, both surgical and endovascular. However, currently there is not strong evidence or codified treatment algorithms supporting which is the best treatment option of CAA in daily practice.

**The OMT**: Antithrombotic and/or anti-ischemic drugs, statins, angiotensin-converting enzyme inhibitors and beta-blockers are the most used drugs in those patients with concomitant atherosclerotic coronary artery disease (CAD) [3,7]. However, currently there are not specific, evidence-based indications for antiplatelet rather than anticoagulant therapy. The CAAR (Coronary Artery Aneurysm Registry) suggests that antiplatelet therapy without additional long-term anticoagulation after revascularization (Percutaneous coronary intervention or coronary artery bypass graft) in acute and non-acute patients might be enough. [23]. Indeed, a sub-analysis of CAAR evaluated 1331 patients who were discharged with or without oral anticoagulation treatment (OAT) with warfarin. At median follow-up of 3 years, the rate of major cardiac event was significantly lower in the OAT group (rehospitalization for unstable angina: 4.6% vs. 10%, *p* < 0.01, aneurysm thrombosis: 0% vs. 3.1%, *p* < 0.03) [24].

In patients with coronary aneurysm who have a recurrence of ACS during dual antiplatelet therapy, the combination of antiplatelet therapy and anticoagulant therapy with warfarin might be considered with INR between 2.0–2.5 [25]. Furthermore, a few case reports described the use of rivaroxaban in CAA, plus aspirin in cases of recurrent angina [26]. Finally, for mycotic aneurisms, an appropriate antibiotic therapy plus aneurism surgery is a cornerstone of therapy [27].

**Invasive therapy.** Depending on the clinical presentation of CAA, or in cases of aneurysms at high risk of complications (e.g., very large CAAs, proximal location), invasive therapy might be considered [3].

**Endovascular treatment.** Most published studies that assessed PCI (Percutaneous coronary intervention) outcomes in patients with CAA have been conducted in the context of acute coronary syndrome (ACS), while very little data exist about patients complaining of chronic coronary artery disease [28,29,30]. The two most used techniques for the treatment of CAA are coil embolization or sealing by covered stent implantation [3]. However, if the aneurysm is very close or involves a bifurcation, the aneurysm sealing might result in side-branch occlusion; a major issue, especially in cases where the side branch is relevant. In such scenarios, the adoption of balloon or stent-assisted coil embolization seems to be effective and safe [7]. In patients with acute coronary syndrome due to a CAA culprit, the emphasis is to restore flow. Due to the higher associated thrombus burden, PCI in ectatic and aneurysmal arteries is frequently aided with thrombectomy (aspiration or mechanical) and glycoprotein IIb/IIIa inhibitors. Despite these efforts, the occurrence of no-reflow or distal embolization is quite frequent [7,28].

Another kind of aneurysm is the aortocoronary saphenous vein graft aneurysm (SVGA). The traditional invasive management for patients with large SVGAs or with symptomatic compression of adjacent cardiac and vascular structures was generally based on the aneurysm resection with or without bypass of the affected territory. Today, besides the surgical treatment, different percutaneous techniques including the use of large plug devices, covered stents and coiling embolization systems are feasible and effective options [31].

**Surgical treatment.** One of the main indications for cardiac surgery is in those patients with CAA and a concomitant heart disease requiring open surgery (e.g., coronary artery bypass graft, relevant valvopathy, ascending aorta aneurysm or multiple giant CCAs). For example, in cases of multivessel disease requiring revascularization and a concomitant relevant coronary aneurysm, this is an important issue that might push the heart team indication toward open surgery over PCI [3,4,5,6,7,8,9,10,11,12,13,14,15,16,17,18,19,20,21,22,23,24,25,26,27,28,29,30,31,32] (Figure 1). Surgical therapy may include aneurysm ligation, resection or marsupialization with interposition graft, and the ideal approach has not yet been formally studied [33].

An analysis of 1565 CAA patients from the international coronary artery aneurysm registry compared long term-outcomes of patients with CAA who underwent CABG (coronary artery bypass graft) or PCI in the setting of acute coronary syndrome (61.5%), stable angina (15.6%) or chest pain (11.2%). The study showed that the number of aneurysms was higher in patients with more severe CAD. At long-term follow (median follow-up was 37.2 months), the endovascular approach was safer and more effective than CABG. Indeed, no differences in major adverse cardiac and cerebrovascular events (MACCE) or death were observed between CABG and PCI groups (respectively, 31.6% vs. 31.4%; *p* = 0.963 and 15.6% vs. 15.9%, *p* = 0.925), while higher rates of heart failure were found in the CABG group (10.8% vs. 5.9%; *p* = 0.009) [23].

In another retrospective study, Khubber et al. compared clinical outcomes of 458 patients with CAA plus coronary artery disease, based on three possible treatment options: OMT, CABG and PCI. At a mean follow-up period of 62 months the total number of MACCE during follow-up was 155 (33.8%); out of those, 91 (39.6%) occurred in the medical management group, 46 (26.1%) in the CABG group and 18 (34.6%) in the PCI group (*p* = 0.02). Kaplan-Meier survival analysis showed that CABG was associated with better MACCE-free survival (p log-rank = 0.03) than medical management [34].

In summary, both endovascular and surgical treatment of CAA been examined with strong levels of evidence, and for this reason it is our opinion that any CAA interventional indication should be shared by the local heart team in a “patient-based fashion”.

## 5. Prognosis

In the past, previous small series have provided conflicting data on the association between coronary artery aneurysms and traditional cardiac risk factors, as well as limited information on patient outcomes. However, in 2004, Baman et al. clarified that CAA might worsen the patient’s long-term prognosis. Indeed, the authors, in a prospective study, enrolled 276 consecutive patients complaining of CA, with and without concomitant atherosclerotic coronary artery disease. It was shown that coronary aneurysms were an independent predictor of mortality. Interestingly, based on whether they had obstructive coronary disease or not, CAA was found to have an adverse effect in five-year follow-up, with a mortality rate of 29.1% (HR 1.65; *p* = 0.77). In light of these results, at that time the authors, reasonably, concluded that patients with CAA should be aggressively followed-up with a concomitant modifying action of coronary risk factors [35].

Another interesting point of discussion is whether CAA might affect acute procedural outcome of those patients undergoing coronary PCI. Yip et al., in a retrospective study, evaluated 24 patients who had an infarct-related artery with aneurysmal dilatation, to verify whether CAA predisposes patients to thrombus formation, no-reflow phenomenon and adverse clinical outcome in patients with acute myocardial infarction undergoing primary PCI.

The reported incidence of cardiogenic shock and the 30-day mortality rate were 25% and 8.3%, respectively; at a mean follow-up of 19 ± 30 months the survival rate was 90.9%. Interestingly, the no-reflow phenomenon and distal embolization rates were 62.5% and 70.8%, respectively [28].

Even if the study cohort is too small for any definitive conclusion, the occurrence of these specific complications seems higher than that reported in other research.

Indeed, previous studies conducted in the same setting, of acute myocardial infarction patients who underwent primary PCI but without coronary aneurysms, reported a no-reflow rate of 31.3% [36] and a distal embolization rate of 15.2% [37].

## 6. Case 1

A 57-year-old man underwent coronary angiography for angina symptoms and inducible ischemia at exercise ECG test. In the patient’s family health history, a case of death due to cerebral aneurysm was reported.

ECG was normal. Echocardiography showed normal size and performance of the left ventricle, mild dilation of the aortic root and proximal ascending aortic aneurysm (48 mm in diameter).

The coronary angiography showed two “in tandem” giant and saccular aneurysms in the proximal and middle left anterior descending artery and another aneurysm in the proximal Circumflex artery. Both LAD and Circumflex artery had a critical stenosis distally to the aneurysms; finally, a right coronary artery chronic total occlusion was shown. (Figure 2A,B).

The thoraco-abdominal MSCTA showed an ascending aortic aneurysm of 48 mm in transversal diameter and a fusiform aneurism of the infra-renal abdominal aorta 30 mm in diameter, with a thrombotic apposition to the aneurysm wall. MSCTA confirmed the presence of a giant aneurysm of the proximal left anterior descending coronary artery that was 20 mm in diameter with a modest thrombotic stratification (Figure 2C–E). The heart team established surgical treatment to be the best treatment option for this patient. A combined coronary artery bypass grafting, surgical exclusion of the coronary aneurysms and ascending aorta replacement with a vascular prothesis was performed without any complications at discharge and at one-month follow-up.

## 7. Case 2

A 51-year-old man underwent emergency coronary angiography for inferior ST-elevation myocardial infarction (STEMI).

The coronary angiography showed a pattern of diffuse ectasia of the right coronary artery with an acute thrombotic occlusion of an aneurismatic posterolateral artery (Figure 3A). A mild and diffuse atherosclerosis affected all the other coronary segments.

A posterolateral artery primary PCI was performed by thrombus aspiration and balloon dilatation. Because of the large thrombus burden, the Eptifibatide was administered and continued up to 18 h after the procedure. The final flow was TIMI 1 (Figure 3B) with a persistent load of thrombus occluding almost completely the distal portion of the vessel.

During hospitalization, the patient underwent dual antiplatelet therapy (aspirin plus ticagrelor). Three days after the index procedure, a repeated coronary angiography was performed. Interestingly, the almost complete recanalization of the distal posterolateral branch was shown with a persistent thin parietal thrombotic stratification at the level of the aneurysm (Figure 3C). The patient was discharged without any complications at discharge and at one-month follow-up.

## 8. Conclusions

CAA affects both patients’ long-term outcomes and the acute procedural outcome in those patients undergoing coronary PCI. The medical, endovascular or surgical treatment of CAA is a challenge today, and in the current clinical practice it is not standardized but rather case-based. Dedicated studies are necessary to definitively establish which is the best medical therapy (antiplatelet, anticoagulant or combined therapy) and when and how CAA should be considered for endovascular or surgical interventions.

## Figures and Tables

**Figure 1 diagnostics-12-02534-f001:**
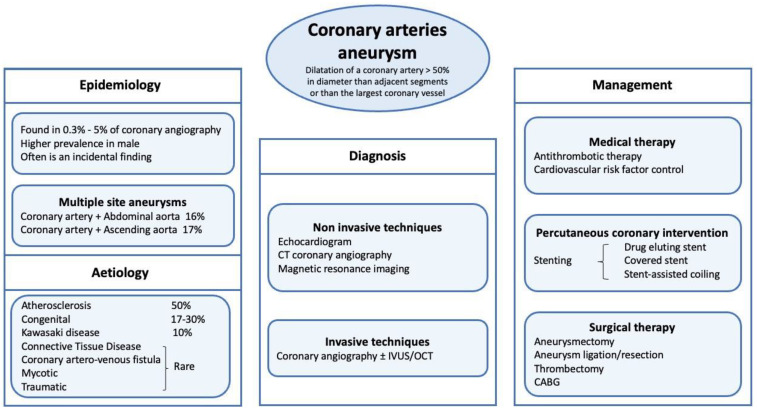
Coronary Artery Aneurysm management. CT (coronary tomography); CABG (coronary artery bypass graft); IVUS (IntraVascular UltraSound); OCT (Optical coherence tomography).

**Figure 2 diagnostics-12-02534-f002:**
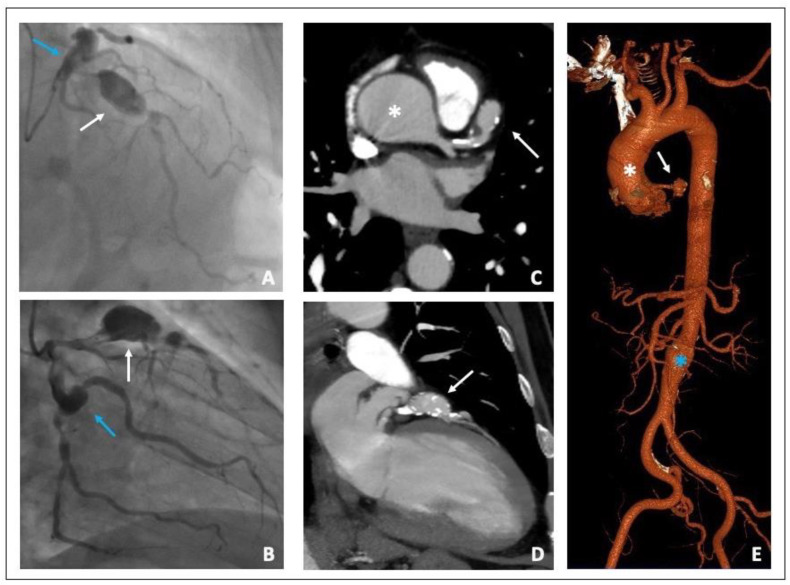
**(A**,**B**) Coronary angiography. (**A**) Proximal LAD giant aneurysm (white arrow) followed by a severe stenosis. Proximal left circumflex artery ectasia (blue arrow). (**B**) Proximal left circumflex artery ectasia (blue arrow) followed by a severe stenosis of the distal segment. Proximal LAD giant aneurysm (white arrow) (**C**,**D**) MSCTA Cardiac short and long axis view showing both ascending aorta (white asterisk) and LAD aneurysms (white arrow). (**E**) 3D aortic reconstruction image: Ascending (white asterisk) and infra-renal aortic aneurysms (blue asterisk) and LAD aneurysm (white arrow). LAD (left anterior descending); MSCTA (multi-slice computed tomography angiography).

**Figure 3 diagnostics-12-02534-f003:**
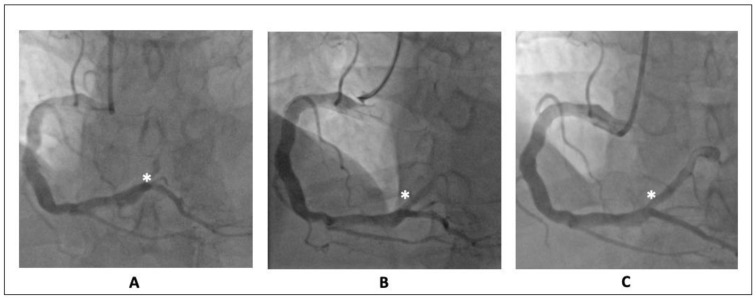
Right coronary artery angiography. (**A**) Diffuse ectasia with a thrombotic occlusion of an aneurismatic posterolateral artery (white asterisk). (**B**) Persistent thrombotic occlusion after balloon dilatation, during eptifibatide infusion (white asterisk). (**C**) Recanalization of the distal posterolateral branch with persistent parietal thrombotic stratification after 3 days from PC (white asterisk).

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
