# Peer review of "Coronary Arteries Aneurysms: A Case-Based Literature Review"

_diagnostics, 2022, doi:10.3390/diagnostics12102534_

Round 1

Reviewer 1 Report

This manuscript presented two cases of atherosclerotic coronary artery aneurysms and had a literature review about giant coronary arteries aneurysms. The authors quite simply discussed the definition, prevalence, etiology, and pathogenesis. We all know that the incidence of giant coronary arteries is relatively low. We still want to be educated with some novel findings and detailed discussions. There are some points for your further improvements.

1. Dual antiplatelet therapy (DAPT) plus anticoagulant (triple therapy) may be used in the high thrombus burden case. You choose a glycoprotein IIb/IIIa inhibitor instead of an anticoagulant in case 2. Please tell us why?

2. What is “CAAR”? Do you mean: coronary artery aneurysm registry?

3. Etiology included atherosclerosis, Kawasaki disease, congenital anomalies, autoimmune diseases, and Marfan’s syndrome. Are there any differences in treatment regarding different etiology of coronary artery aneurysms? You may consider below case report for reference (Catheter Cardiovasc Interv. 2000 Nov;51(3):328-31).

Author Response

Dear reviewers,

Dear editor,

Thank you for reviewing our manuscript entitled “Giant coronary arteries aneurysms: a case-based literature review” (Manuscript ID: diagnostics-1960183).

Reviewer 2 Report

I would like to thank the authors for the chance of reviewing their article "Giant coronary arteries aneurysms: a case-based literature review". Vadala et al provided an updated review on the coronary arteries aneurysms and they provided the imaging for two of their cases in support.

Before proceeding further, I would like to address some issues:

- As per reference no.3 "Diameters greater than 20 mm, 40 mm or 50 mm, or quadruple the reference vessel diameter have all been proposed as a definition in the medical literature, but there is still no clear consensus on how a GCAA should be defined" and as per reference no. 7 "The term “giant” aneurysm is applied when the size exceeds 5 cm in diameter; these giant aneurysms are usually saccular "- either a reconsideration of the cases used in this article could be of interest or a different definition of "giant".

- Regarding Case 2 in the current paper: From the images it does not seems to be a giant coronary aneurysm (as per the title of the article). Maybe a reconsideration of either the title or the inclusion of a different case would be worth considering.

- This article is about coronary artery aneurysms, and the title is misleading. Either the title and abstract should be re-written or more cases should be included.

Author Response

(The authors gave the same response as above.)

Round 2

Reviewer 1 Report

I personally agree with this manuscript after revision. 

Reviewer 2 Report

Thank you for the updated manuscript; I have no further comments.